# Effects of Seawater Polyphenols from *Gongolaria usneoides* on Photosynthesis and Biochemical Compounds of the Invasive Alien Species *Rugulopteryx okamurae* (Phaeophyceae, Heterokontophyta)

**DOI:** 10.3390/plants14162594

**Published:** 2025-08-20

**Authors:** Débora Tomazi Pereira, Fernando García Alarcón, Manolo García Alarcón, Paula S. M. Celis-Plá, Félix L. Figueroa

**Affiliations:** 1Experimental Center Grice Hutchinson, University Institute of Blue Biotechnology and Development (IBYDA), University of Malaga, Lomas de San Julián, 2, 29004 Málaga, Spain; felixlfigueroa@uma.es; 2Equilibrio Marino Association, Calle Honduras, 9, 1ºB, 29007 Malaga, Spain; info@equilibriomarino.com (F.G.A.); lolo@equilibriomarino.com (M.G.A.); 3Laboratory of Aquatic Environmental Research (LACER)/HUB-AMBIENTAL UPLA, Playa Ancha University, Valparaíso 2360004, Chile; paulacelispla@upla.cl; 4Departamento de Ciencias y Geografía, Facultad de Ciencias Naturales y Exactas, Universidad de Playa Ancha, Valparaíso 2360004, Chile

**Keywords:** alga, environmental, invasive species, phenolic compounds, photosynthesis, recovery, toxicity

## Abstract

*Gongolaria usneoides* is a native brown alga in Europa, known for its high release of phenolics into the water. *Rugulopteryx okamurae* is a native brown macroalga from Asia but invasive in Europe, causing significant environmental and socioeconomic impacts. It has been observed that in some regions where *Gongolaria* is present, there is less biomass of *Rugulopteryx*, and they are often epiphytized by red filamentous algae. The present study aimed to analyze whether phenolics released in the seawater by *G. usneoides* affect the photosynthetic and biochemical responses in *R. okamurae*. To analyze the resilience of *R. okamurae*, algae were cultivated for 5 days in water with different percentages of phenolics released by *G. usneoides* (exposure phase) (0, 75, 150, 225 and 300 µg mL^−1^) in laboratory (20 °C, a 12:12 photoperiod, and PAR 190 µmol photons m^−2^ s^−1^ for a period of 115 h) and a period in the sun (4 h), followed 2 days cultured under phenolic-free water (recovery phase). Photosynthetic characteristics (*Fv*/*Fm* and ETR) and biochemical composition (phenolic, antioxidant activity, C, N and S) were measured (*R. okamurae* showed considerable photosynthetic declines without recovery when exposed to high concentrations of phenolics (225 and 300 µg mL^−1^). It can be inferred that the presence of *G. usneoides* in nature and the release of phenolic compounds by this alga may be affecting the invasive alga *R. okamurae*, potentially serving as a natural means to decrease or weaken the invasive species.

## 1. Introduction

Macroalgae are ecologically essential organisms as they are primary producers, performing photosynthesis, thus playing a significant ecological role by providing oxygen for the respiration of other living organisms, as well as habitat and feed [1]. On the other hand, when macroalgae species become invasive in a particular region, these organisms start to bring harm, as biological invaders are considered one of the most important drivers of biodiversity loss, with ecological and economic impacts [2,3,4,5,6,7]. In this way, maritime transportation, both through ballast waters and hull fouling, has been identified as a significant pathway for the introduction of species into new environments [8]. The Strait of Gibraltar presents the highest density of maritime traffic in the Western Mediterranean [9,10].

*Rugulopteryx okamurae* (E.Y. Dawson) I.K. Hwang, W.J. Lee & H. S. Kim is a native brown macroalgae from East Asia (China, Japan, Korea). The presence of this species was reported in Thau Lagoon (France) in 2002 associated with Japan oyster culture [11]. However, this species was first recorded as an Invasive Alien Species (IAS) in the open sea of the Mediterranean at the Strait of Gibraltar (Ceuta, Spanish Peninsula, and Morocco) in 2015 [12,13,14], where it quickly became invasive. The massive expansion of *R. okamurae* has caused significant negative impacts, both environmentally, leading to a considerable loss of native biodiversity [15,16], and socioeconomically, affecting the fishing sector as well as tourism due to the high accumulation of algal biomass in floating and coastal forms [13,17,18,19,20,21]. Due to all these characteristics, in 2020, the Ministry of Ecological Transition and Demographic Challenge added *R. okamurae* to the Spanish Catalogue of Invasive Alien Species.

*Gongolaria* is a genus of brown macroalgae primarily found along the coasts of the Baltic Sea, south of the Mediterranean Sea, the Canary Islands, Mauritania, and Western Sahara [22]. They are algae of significant importance as they are considered the climax community of rocky shores in the Mediterranean [23,24,25,26,27].

Brown macroalgae can release phenolic compounds, causing a yellowish-brownish color in seawater or in culture media, and these compounds have been suggested as contributors to ‘gelbstoff’ in seawater [28,29,30,31]. This release is associated with stressful conditions, such as elevated levels of irradiance and temperature conditions [30,32,33]. Phenolic compounds are known to function as protectors against UV radiation, antioxidants, anti-herbivore defense, resistance to pathogens, and defense against epiphyte growth [34]. Swanson & Druehl [35] showed that kelp *Macrocystis integrifolia* phlorotannin exudates can affect seawater UV transparency and the internal phlorotannin levels in the short-term.

Phenolic compounds are among the most abundant and widely distributed groups of secondary metabolites in nature [36]. They are essential for the normal development of both algae and terrestrial plants, contributing to defense mechanisms against infections, physical damage, and a variety of environmental stressors [36,37]. These compounds play critical roles in physiological processes, including protection against UV radiation, herbivory, and pathogens, as well as the inhibition of epiphyte colonization [34,36,38]. Moreover, phenolic compounds exuded by brown macroalgae into the surrounding seawater can act as chemical signals or allelochemicals, influencing the behavior and growth of nearby organisms such as microalgae and herbivores [34,39].

Although *Cystoseira*-*Gongolaria* meadows are being affected by *R. okamurae*, it is observed that the community presents a certain resistance to the invasion and it is not completely disturbed, as occurred with other species in the Strait of Gibraltar water [16]. One of the possible mechanisms of resistance is the exudation of phenolic compounds, among other compounds, as a response to the presence of *R. okamurae.* Celis-Plá et al. [40,41,42,43] shown an increase in polyphenol content and antioxidant capacity in different *Cystoseira* species in the Mediterranean and Atlantic Ocean in response to different stressors, such as UVR, acidification, heavy metals or increased temperature.

Therefore, the present study aimed to analyze whether the presence of phenolic compounds released in the seawater by *Gongolaria usneoides* has effects on the photosynthetic and biochemical response of *R. okamurae*, and how these algae recover after transferring to optimal conditions, in an attempt to find a natural chemical defense means to decrease/weaken the invasive species [44].

## 2. Results

### 2.1. Toxicity and Recovery Tests

The phenolic release from *G. usneoides* was conducted in the laboratory, resulting in a final concentration of 300 µg mL^−1^. Phenolic release in natural tide pools by *G. usneoides* reached approximately 18–50 µg mL^−1^ after 2 h of sun exposure. Under controlled laboratory conditions (190 µmol photons m^−2^ s^−1^ and 20 °C), the natural release of phenolics after 2 days was around 150 µg mL^−1^. Although this is a low value compared to the experimental conditions, detecting all phenolic compounds released into the water becomes challenging in open environments. Subsequently, the original phenolic solution was mixed with artificial seawater to obtain different phenolic concentrations (Figure 1). The phenolic compounds’ spectra showed phloroglucinol, according to the commercial standard, with a peak of absorption of 270 nm.

#### 2.1.1. Optimal Quantum Yield (*F_v_*/*F_m_*) and Electron Transport Rate (ETR)

In the laboratory under artificial white light exposure, the algae *R. okamurae* were exposed to different phenolic concentrations for 115 h under controlled conditions (20 °C, a 12:12 photoperiod, and PAR 190 µmol photons m^−2^ s^−1^). Due to the coloration of polyphenols in water, the amount of PAR reaching the center of the Erlenmeyer flask varied, justifying the difference between the electron transport rate (ETR) observed at the end of this period (Table 1).

For the epiphytic algae, at the end of the laboratory period (115 h), the optimal quantum yield *(F_v_*/*F_m)_* was highest in the sample with 25% phenolic concentration, and lowest at 100% (Table 1, Figure 2a). Then, the algae were transferred for 4 h of exposure to polyphenols under solar radiation (119 h); the highest *F_v_*/*F_m_* was detected in samples exposed to 0, 25 and 50%, while the lowest *F_v_*/*F_m_* was in those exposed to 75 and 100% (Figure 2a). Finally, at 24 h and at the end of recovery test, the algae continued to follow the same pattern observed at the end of the solar exposure (Figure 2b).

Considering normal *R. okamurae*, at the end of the laboratory period, the *Fv*/*Fm* was lowest at 100% phenolic concentration, while it was highest for all other treatments (Table 1, Figure 2c). After 4 h of exposure to polyphenols under sunlight, the samples continued to follow the same pattern observed at the end of the laboratory exposure (Figure 2c). At 24 h, as well as at 48 h after starting the recovery test, the highest *F_v_*/*F_m_* was detected in samples exposed to 0, 25 and 50%, while the lowest *F_v_*/*F_m_* was detected in those exposed to 75 and 100% of phenolic concentration (Figure 2d).

The slope of the line, both for epiphytic and normal algae, and for all concentrations of phenolics in water, was negative; however, higher slopes were identified in the treatments with 75% and 100% phenolics (Figure 2b,d).

After sun exposure, in epiphytic *R. okamurae*, the ETR under all phenolic concentrations and solar exposure did not show significant differences. At 1 h of the recovery test, the samples exposed to 0% phenolic concentration exhibited the lowest ETR, while in all the other samples, no statistical differences were observed. On the other hand, at 3 and 24 h of recovery, the samples exposed to 100% phenolic concentration showed the lowest ETR, while no statistical differences were detected in the other treatments. After 48 h of recovery, the samples exposed to 75 and 100% phenolic concentrations exhibited the lowest ETR among all other treatments (Figure 3a).

Considering the normal *R. okamurae*, during phenolic and solar exposure, significant differences were observed only at 2 h, where samples exposed to 75% showed the lowest ETR. At 1 h of the recovery test, there was no statistically significant difference among all the treatments. However, at 3 and 24 h of recovery, the samples exposed to 50, 75 and 100% phenolic concentrations exhibited the lowest ETR, particularly those exposed to 100% concentration. At 48 h of recovery, only samples exposed to 75 and 100% phenolic concentrations showed the lowest ETR among all other treatments (Figure 3b).

Calculating the percentage of ETR drop in relation to the phenolic concentration, a positive and strong Pearson’s correlation was observed in epiphytic algae (r = 0.7156, *p* = 0.003) and in normal algae (r = 0.8601, *p* = 0.0001). A similar positive and strong Pearson’s correlation was detected in the percentage of *Fv*/*Fm* drop with phenolic concentration in both epiphytic (r = 0.7720, *p* = 0.001) and normal algae (r = 0.8069, *p* = 0.0001).

#### 2.1.2. Polyphenols and Antioxidant Activity

The internal phenolic content in epiphytic *R. okamurae* after the recovery test did not show statistically significant differences between samples exposed to 0, 25, 50 and 75% of phenolic concentration. However, a significant decrease was observed in the sample exposed to 100% phenolic concentration. For normal *R. okamurae,* after the recovery test, no treatment showed statistical differences. When analyzing all samples (phenolic concentration and *R. okamurae* condition), it was detected that samples exposed to 100% phenolic concentration showed the lowest internal phenolic content. The percentage of phenolics in water was the factor that had the greatest effect on the internal concentration of phenolics in the algae [η^2^ = 46.03%, F_(4,20)_ = 5.3965, *p* ≤ 0.004]. No statistical differences were observed regarding samples treated with the same phenolic concentration (Figure 4a).

After the recovery test, no samples, whether only epiphytic, normal, or all combined, showed any statistically significant differences. Even without showing statistical differences, the percentage of polyphenols in water was the factor that had the greatest effect on the antioxidant activity in the algae [η^2^ = 54.01%, F_(4,20)_ = 6.8384, *p* ≤ 0.001]. No statistical differences were observed regarding samples treated with the same phenolic concentration (Figure 4b).

#### 2.1.3. Internal Carbon, Nitrogen, and Sulfur

The internal carbon in both epiphytic and normal *R. okamurae* after the recovery test did not show any statistical differences among all the treatments. Even without showing statistical differences, the percentage of phenolics in water was the factor that had the greatest effect on the internal carbon content in the algae [η^2^ = 46.69%, F_(4,20)_ = 5.92, *p* ≤ 0.002]. When analyzing all samples, including different phenolic concentrations and *R. okamurae* conditions, it was detected that epiphytic samples exposed to 100% phenolic concentration showed the lowest internal phenolic content. No statistical differences were observed regarding samples treated with the same phenolic concentration (Figure 5a).

After the recovery test, in epiphytic *R. okamurae,* the lowest internal nitrogen content was detected in samples exposed to 0% phenolic concentration, while the other phenolic treatments did not show any statistical difference. Normal *R. okamurae* exposed to 0% phenolic concentration showed the lowest internal nitrogen content, followed by samples exposed to 25, 50 and 100% phenolic concentration, with the sample exposed to 75% having the highest in internal nitrogen content. When analyzing all samples, including different phenolic concentrations and *R. okamurae* conditions, it was detected that both epiphytic and normal samples exposed to 0% phenolic concentration showed the lowest internal nitrogen content, whereas normal samples exposed to 50 and 75% and epiphytic samples exposed to 50% phenolic concentration showed the highest nitrogen content. The percentage of polyphenols in water was the factor that had the greatest effect on the internal nitrogen content in the algae [η^2^ = 85.35%, F_(4,20)_ = 72.28, *p* ≤ 0.001]. No statistical differences were observed regarding samples treated with the same phenolic concentration (Figure 5b).

At the end of the recovery test in epiphytic *R. okamurae,* the highest internal sulfur content was detected in samples exposed to 0% phenolic concentration, while the other phenolic treatments did not show any statistical difference. Normal *R. okamurae* exposed to 0 and 100% phenolic concentrations showed the lowest internal nitrogen content, while samples exposed to 50% phenolic concentration had the highest internal sulfur. When analyzing all samples, including different phenolic concentrations and *R. okamurae* conditions, it was detected that epiphytic samples exposed to 0% phenolic concentration showed the highest internal sulfur content, while normal samples exposed to 0 and 100%, as well as epiphytic samples exposed to 50, 75 and 100%, exhibited the lowest sulfur content. The interaction between the factors (phenolic concentration and *R. okamurae* condition) demonstrated a more substantial effect on the internal sulfur content in the algae [η^2^ = 68.77%, F_(4,20)_ = 51.789, *p* ≤ 0.001]. Upon analyzing epiphytic and normal samples treated with the same phenolic concentration, statistical differences were observed between 0% and 100% phenolic samples, denoted by asterisks (Figure 5c).

At the end of the recovery test, the carbon to nitrogen ratio (C:N) in both epiphytic and normal algae showed the highest values in samples exposed to 0% phenolic concentration, and the lowest values at 100% for epiphytic and 75% for normal algae (Table 2).

Regarding the carbon to sulfur ratio (C:S), the highest C:S ratio for epiphytic algae was observed in samples exposed to 50%, while the lowest was in those exposed to 0%. For normal algae, the highest C:S ratio was detected in samples exposed to 0%, and the lowest in those exposed to 50% phenolic concentrations (Table 2).

Finally, for the nitrogen to sulfur ratio (N:S), the highest ratio for epiphytic algae was observed in the sample exposed to 50%, while for the normal algae, the highest was observed in sample exposed to 100% phenolic concentration. The lowest value was detected in epiphytic algae exposed to 0%, and for normal algae, it was reached at a 50% phenolic concentration (Table 2).

## 3. Discussion

The *G. usneoides* presents intra-thallus variation in polyphenol and epiphytic density. The apical zone exhibited more epiphytic algae than the intermediate and basal zones. It is known that the apical zone is more sensitive than the basal zone, possessing compounds such as phenolics/phlorotannins that can be toxic to other organisms, providing effective chemical defenses against certain marine vertebrate/invertebrate herbivores and epiphytism [30,45,46,47], and defense against epiphytic algae and bacteria [31]. Indeed, Karban & Baldwin [48] is thought to be an indirect effect of excreted phlorotannins in algae which are released into the water when algae are grazed. In this study, the content of internal content polyphenols in *G. usneoides* in the different parts was negatively correlated with the amount of epiphytes in the thalli, i.e., basal did not present epiphytes, being the part with the highest content, whereas the highest content of epiphytes was observed in the apical part with the lowest level of internal polyphenols. It is expected that excreted polyphenols produce chemical control against epiphytism. Intra-thallus variation in phenolic compounds has also been reported in *Cystoseira tamariscifolia,* with the highest amount in the apical part correlated with the highest antioxidant capacity and phenosulphatase activity related to phenolic excretion [49]. Regarding the two types of *R. okamurae*, one was more reddish, and the normal specimens were more brownish, which is the typical color of the species. The more reddish alga, located near the zone with *G. usneoides*, exhibited this coloration because it is highly epiphytized, especially by red algae, making it more vulnerable and sensitive to being epiphytized. By contrast, *R. okamurae* located out of *G. usneoides* meadows is brownish because the epiphytes are scarce and the cover on the sea bottom is close to 100%. In contrast, epiphytized algae had much lower density compared to normal specimens.

Regarding the amount of internal phenolics, *G. usneoides* exhibits concentrations much higher than those of *R. okamurae*. It is due to this large production of internal phenolics that this species can externalize and release these compounds as a possible defense, signaling and protection mechanism [30,49]. In the case of *Gongolaria*, the literature reports that the most abundant phenolic compound is benzoic acid [50], whereas in *R. okamurae*, benzoic acid is not present in its composition, and the predominant phenolic compound is gallic acid [51]. As these are antioxidant compounds, one of their main functions is protection against oxidative stress [48]. High PAR irradiances and emersion have been associated with increasing phlorotannin release rates [29,52], and in the same way, the stimulation of polyphenol production by UVR exposure has been previously reported in *Gongolaria*/*Cystoseira* species [30,32]. Indeed, Celis-Plá et al. [30] also found a higher release rate of polyphenols from *C. tamariscifolia* in outdoor experiments in the summer or higher PAR + UVR compared to winter or lower PAR + UVR. In consequence, the release of phenolic compounds is thought to be a photoprotection mechanism due to the transient reduction in UV penetration favored by the accumulation of excreted phenols in the cell wall. Furthermore, *G. usneoides* serves as a habitat and refuge for small marine animals, and it also functions as a food source for fish and other marine organisms [53,54]. The movement of animals around the alga, as well as the breaking of its thallus due to bites and mechanical interactions, are important factors contributing to the release of phenolic compounds into the surrounding water.

Regarding the toxicity test of phenolics released by *G. usneoides* on the two ‘types’ of *R. okamurae*, it was noted that the ETR decreased after 2 h of sun exposure. The ETR calculation is directly related to the amount of PAR. On the day of solar exposure, it was transiently cloudy, which reduced the amount of PAR during this time. However, shortly after, the PAR amount increased, causing the ETR to increase as well. For both the ‘normal’ and epiphytized *R. okamurae*, a statistical difference in ETR was observed after 3 and 24 h of recovery in algae exposed to 100% phenolics. At the end of the recovery period (48 h), samples from the 75% and 100% phenolic exposure showed the lowest ETR, indicating that exposure to high concentrations of phenolics affected the photosynthetic apparatus of these algae. The decreased photosynthetic capacity under the highest level of polyphenols in water (75 and 100%) can be explained by pro-oxidant effects of polyphenols [55].

The *F_v_*/*F_m_* ratio indicates the maximum quantum yield of an alga related the physiological state of photosynthetic organisms and photoinhibition [56]. For both types of *R. okamurae*, the exposure period to phenolic concentrations in the culture room, under controlled conditions, showed that samples with 100% phenolics had the lowest *F_v_*/*F_m_*. Algae in locations with lower light availability tend to increase their *F_v_*/*F_m_*, which was not observed during this laboratory exposure period. This suggests that the decrease in *F_v_*/*F_m_* was not related to the decrease in PAR irradiance produced by dissolved polyphenols but rather by the bioactivity of phenols against physiological activity in *R. okamurae*. Thus, phenols produce a stress condition on photosynthesis causing photoinhibition, as well as other stress conditions such as high temperature, UVR, acidification or cupper [30,32,43].

After exposure to phenolics under sunlight with higher irradiance than that in the laboratory, the epiphytized *R. okamurae* showed lower *Fv*/*Fm* in algae exposed to 75% and 100% phenolics, while the ‘normal’ algae had the lowest *Fv*/*Fm* only when exposed to 100% phenolics. After the recovery period, for both types of *R. okamurae*, the same pattern observed in ETR was noted; algae initially exposed to 75% and 100% phenolics did not recover their photosynthetic apparatus, showing the lowest *Fv*/*Fm* rates. All these results are corroborated by Pearson’s correlation, where higher concentrations of phenolics in the water correspond to greater declines in ETR and *Fv*/*Fm* after the recovery period.

Internal phenolics are antioxidant compounds used for the defense of algae and can also have pro-oxidant activity [53,54]. For the two types of *R. okamurae* analyzed in this study, it was observed that when algae were subjected to higher percentages of phenolics in the water, even after the recovery test in the absence of phenolics in the water, lower concentrations of internal phenolics were observed. This indicates that the high initial presence of phenolics in the water caused a utilization or release of the internal phenolics in *R. okamurae*, which can be associated with an effect on the defense metabolism of these algae. However, the antioxidant activity of *R. okamurae* did not show statistically significant differences in relation to the different treatments (amount of phenolics in the water), nor between the two types of *R. okamurae*, possibly associated with the presence of other antioxidants not quantified in this study, such as fucoxanthin [57,58,59].

After exposure to phenolic concentrations and the recovery test, *R. okamurae* did not show statistical differences in the amount of internal carbon either between treatments or between the ‘types’ of algae, indicating that even though the cell was weakened, there was no total cellular degradation during this exposure period. Regarding the amount of internal nitrogen, the lowest concentrations were observed in the two types of *R. okamurae* exposed to 0% polyphenols in water. Although not measured in this study, it is known that cellular defense proteins containing nitrogen, such as antioxidant enzymes like catalase, superoxide dismutase, and peroxidases, may be increased under stress conditions with the presence of phenolics in the water [60]. As for the total sulfur, the epiphytized algae demonstrated the highest amount in algae exposed to 0% phenolics in water, while the ‘normal’ algae showed the highest amount of sulfur when exposed to 50% phenolics in water.

## 4. Materials and Methods

### 4.1. Collection of Biological Material

Specimens of macroalgae *Gongolaria usneoides* (Linnaeus) Molinari & Guiry and *Rugulopteryx okamurae* (E.Y.Dawson) I.K.Hwang, W.J.Lee & H.S.Kim (Phaeophyceae, Heterokontophyta) were collected at the coast of Estepona, Spain (36°27′ N and 4°58′ W), at 14 m depth (Figure 6). Algal thalli were transported to the Institute of Blue Biotechnology and Development (IBYDA) of Malaga University (Spain) in plastic containers filled with seawater and placed inside a thermal box. In the laboratory, the thalli were rinsed with diluted artificial seawater (13 psu, Instant Ocean^®^, Blacksburg, VA, USA).

### 4.2. Characterization of Biological Material

For microscopic characterization, fresh material was spread in a Petri dish with water and observed under a stereoscopic microscope (Leica, Wetzlar, Germany) equipped with a Leica ICC50 W camera. The software LAS-EZ (version 3.4, Leica, Wetzlar, Germany) was used for final image processing.

The brown algae *G. usneoides* measured 125 cm in length. They were divided into three zones: apical, intermediate, and basal. Microscopically, the apical zone exhibited 6 epiphytic species: *R. okamurae*, *Aglaothamnion* sp., *Ceramium cf secundatum*, *Ceramium* sp1., *Crouania attenuata*, and *Fankenbergia* (=*Asparagopsis*, tetraesporophyte). The intermediate zone had only one epiphytic species, *Ceramium* sp1., while the basal zone was not epiphytized (Figure 7).

When observed under a stereoscopic microscope, it was noticed that *R. okamurae,* with a reddish coloration, found in the same area as *G. usneoides*, was epiphytized with 8 the following algae: *Aglaothamnion* sp., *Antithamnionella* sp., *Ceramium* sp1., *Crouania* sp., *Cyanophya* sp., *Gayllela* sp., *Hypoglosum* sp., and *Polysiphonia* sp. (Figure 8a,c). Meanwhile, the *R. okamurae* found in the area without *G. usneoides* remained free or with few epiphytes (*Cyanophyta* sp. and *Gastroclonium*/*Chylocladia* sp.), maintaining its typical brown coloration characteristic of this species (Figure 8b,d).

The internal phenolic concentration in *G. usneoides* was highest in the basal zone when compared across all zones, while the apical and intermediate zones showed no statistically significant difference between them. In *R. okamurae*, there was no statistical difference between the ‘epiphytized’ algae and the normal (non-epiphytized) individuals. Moreover, the internal phenolic content in *R. okamurae* was significantly lower compared to that of *G. usneoides* (Table 3).

### 4.3. Obtaining Signaled Water and Exudation of Phenolic Compounds

For the release of phenolic compounds, 400 g of fresh *G. usneoides* was mashed with 4 L of artificial seawater (Instant Ocean^®^) at a salinity of 36 psu. This mixture was allowed to macerate for 1.5 h in a thermostatically controlled water bath set at 80 °C. Afterward, the resulting extract was subjected to vacuum filtration and subsequently utilized in the toxicity test.

The release of phenolics into the seawater was determined by measuring the optical density at 270 nm as the maximum absorption wavelength in seawater (according to phloroglucinol as commercial standard 1,3,5-trihydroxybenzene, Sigma P-3502, St. Louis, MO, USA) using a UV–visible spectrophotometer (Shimadzu UV-2600, Kyoto, Japan) according to Celis-Plá et al. [30]. The concentration was expressed as μg.mL^−1^ and was determined using a standard curve of phloroglucinol (4 to 80 µg mL^−1^ − r^2^ = 0.99, y = 0.0150x + 0.0088).

### 4.4. Toxicity and Recovery Tests

The toxicity test involved the use of 250 mL Erlenmeyer flasks containing 600 mg of *R. okamurae* (*n* = 3). To establish distinct phenolic concentration levels, five gradients were formulated: 0%, 25%, 50%, 75%, and 100% phenolic concentration in artificial seawater (Instant Ocean^®^) at a salinity of 36 psu. Exposures were conducted in two stages: under laboratory conditions (photosynthetically active radiation (PAR) of 190 µmol photons m^−2^ s^−1^ (White LED light, 5000 K, 54 W, NU-8416, Nuovo, Malaga, Spain), temperature of 20  ±  2 °C, a 12 h photoperiod, and continuous aeration) for a period of 4.5 days, and under solar irradiance for a period of 4 h (from 10:00 to 14:00), where mesh screens were employed to standardize PAR exposure across all samples. This standardization was necessary due to light attenuation caused by the presence of elevated phenolic concentrations, which reduced radiation penetration within the flask. Measurements of photosystem II yield (Y_II_) ((Diving-PAM II, Walz^®^, Effeltrich, Germany) with red light provided by light-emitting diodes as measuring, actinic and saturation light pulse irradiation) and the intensities of PAR (LI-250A, LI-COR Biosciences, Lincoln, Nebraska, NE, USA) were recorded at the end of the sun exposure in the thalli submerged in the chambers (in situ measurements). After the 4 h exposure period, the samples were subjected to a recovery test conducted within a controlled cultivation chamber set at 20 °C with a PAR irradiance of 190 µmol photons m^−2^ s^−1^. During this phase, the phenolic-laden aqueous milieu was replaced by 250 mL of artificial saline water at 36 psu. Subsequently, Y_II_ was taken at intervals of 1 h, 3 h, 24 h, and 48 h post-commencement of the recovery phase.

The duration of the experimental phases was determined based on photosynthetic measurements. Once a significant physiological response was detected, the recovery phase was initiated. This phase lasted 48 h, simulating a short-term response that reflects natural conditions, where macroalgae typically stop exuding phenolic compounds into the water for only brief periods.

Quantification of phenolic content and antioxidant activity was performed using freshly collected samples at the commencement of the toxicity experimentation and after the recovery test. An experimental design is shown in Figure 9 to facilitate understanding of all the steps.

### 4.5. In Situ Measurements of Effective Quantum Yield of PSII

Measures of the effective quantum yield of PSII (*Y_II_*) were conducted in the thalli submerged in the chambers by using a portable pulse amplitude modulated fluorometer (Diving-PAM II) at the end of a 4 h sun exposure (from 10:00 to 14:00), and at 1, 3, 24 and 48 h after the start of the recovery test. The *Y_II_* was used to calculate ETR as follows [61]:ETR (µmol photons e^−^ m^−2^ s^−1^) = *Y_II_ × E_PAR_ × A × F_II_*(1)
where *Y_II_* is the effective quantum yield, and *E_PAR_* is the incident PAR irradiance expressed in µmol photons m^−2^ s^−1^. Irradiance of PAR was determined by using quantum cosine corrected sensor (Licor 190SA) connected to a Radiometer Licor-250A (Licor, Lincoln, NE, USA). *A* is the thallus absorptance as the fraction of incident irradiance that is absorbed by the algae [62] and *F_II_* is the fraction of chlorophyll related to PSII (400 e 700 nm) being 0.80 in brown macroalgae [63]. *F_o_* and *F_m_* were measured after 15 min in darkness to obtain the maximum quantum yield (*F_v_*/*F_m_*) where *F_v_ = F_m_ − F_o_. F_o_* is the basal fluorescence of 15 min dark adapted thalli and *F_m_* is the maximal fluorescence after a saturation light pulse of >4000 µmol photons m^−2^ s^−1^ [61]. A linear regression slope (m) was calculated for the %*Fv*/*Fm* and %ETR data across the entire experimental period to evaluate the overall trend in physiological response.

### 4.6. Total Phenols

The phenolic compounds were extracted from 200 mg of fresh algae with 5 mL sodium phosphate buffer (0.1 M, pH 6.5). This extraction process was facilitated using an UltraTurrax^®^ (T25, IKA, Berlin, Germany) (18,000 rpm), followed by a 24 h incubation period at a temperature of 4 °C.

The analysis of phenolic compounds was conducted using the spectrophotometric Folin–Ciocalteu method based on the work of Folin & Ciocalteu [64]. Aliquots of 50 µL from the algal extracts were added to 750 µL distilled water, 50 µL of the Folin–Ciocalteu reagent (Sigma-Aldrich, St. Louis, MO, USA), and 150 µL of 20% sodium carbonate, and incubated for 2 h at 4 °C. Subsequently, readings were taken at 760 nm using a UV–visible spectrophotometer. The quantification of total phenolic compounds was determined using a standard curve of phloroglucinol (1 to 20 µg mL^−1^ − r^2^ = 0.99, y = 0.0586x − 0.0003). The analysis was performed in triplicate, and the results were expressed in mg of phloroglucinol per g of DW.

### 4.7. Total Antioxidant Activity

Antioxidant capacity was determined by the ABTS radical scavenging assay. Firstly, the radical cation ABTS^+•^ was prepared by mixing 7 mM of ABTS (2,2′-azino-bis (3-ethylbenzothiazoline-6-sulphonic acid, 7 mM)) (Sigma-Aldrich) and 2.45 mM of potassium persulfate (K_2_S_2_O_8_) in a sodium phosphate-buffered solution (0.1 M, pH 6.5). The mixture was incubated in darkness at room temperature for 16 h for the complete formation of the radical. For the reaction, ABTS^+•^ was diluted with phosphate buffer until absorbance at 727 nm were 0.75 ± 0.05. The assay was carried out by adding 950 μL of the diluted ABTS^+•^ solution and 50 μL of algal extract, according to Re et al. [65]. The samples were shaken, and absorbance was recorded by a UV–visible spectrophotometer at 727 nm at the beginning of the reaction (DO_i_) and after 8 min of incubation (DO_f_). The percentage of antioxidant activity (AA%) was calculated according to the following formula:AA% = [(absDO_i_ − absDO_f_)/absDO_i_] × 100(2)

The antioxidant compounds concentration was calculated using a standard curve of Trolox (6-hydroxy-2,5,7,8-tetramethylchroman-2-carboxylic acid) (Sigma-Aldrich) (20 to 100 µg mL^−1^ − r^2^ = 0.99, y = 15.893x + 2.7335), and the results were expressed in µmol of Trolox equivalent antioxidant capacity (TEAC) per g of DW.

### 4.8. Total Internal Carbon, Nitrogen, and Sulfur

Total internal carbon (C), nitrogen (N), and sulfur (S) contents of the dry algal biomass were evaluated using a LECO-932 CNHS elemental analyzer (LECO, St. Joseph, MI, USA) in the Research Support Central Services (SCAI, University of Malaga, Spain). The analyses were performed in triplicate, and the results were expressed in %.

### 4.9. Statistical Analysis

The data passed the Shapiro–Wilk’s normality test and the Bartlett test for homogeneity of variances, and all samples were within the normal range and exhibited homoscedasticity. Afterwards, the data were analyzed by unifactorial and bifactorial analysis of variance (ANOVA) and Tukey’s a posteriori test (*p* ≤ 0.05), and, in some cases, the independent-sample *t*-test was performed (*p* ≤ 0.05). Correlations were obtained by Pearson’s correlation coefficient in bivariate correlations. Statistical analyses were performed using the Statistica software package (Release 10.0).

## 5. Conclusions

This study shows the effects of polyphenols in seawater from *G. usneoides* on the photosynthetic and biochemical composition of *R. okamurae*. Despite the higher concentrations used in the laboratory experiment being lower than those measured in the field, the results indicate a research line for further experiments to understand the interactions among macroalgae. *Posidonia oceanica,* a Mediterranean marine angiosperm that is resisting the invasion by *R. okamurae* compared to a great number of macroalgal species, also presents a high content of polyphenols and high diversity [66,67,68]. *P. oceanica* is in interaction with the green macroalga *Caulerpa taxifolia*, it accelerates the production of polyphenols, being secondary metabolites to limit invasion of the beds [67]. The release of *P. ocenaica* phenols could also be contributing to the defense against *R. okamurae,* as we proposed in this study with *G. usneoides*.

Based on the results obtained in this study, it can be concluded that *G. usneoides* presents a higher amount of internal phenolic compounds than that of *R. okamurae*, providing greater antioxidant activity compared to the exotic brown algae. It was also noted that whether *R. okamurae* is epiphytized or not does not cause differences in internal phenolic compounds. Furthermore, it was concluded that after exposure to different concentrations of phenolics derived from *G. usneoides* in water, both types of *R. okamurae* (epiphytized and normal) showed considerable photosynthetic declines without recovery when exposed to high concentrations of phenolics (75% and 100%) and the species of epiphytic algae on *R. okamurae*, as photosynthesis was measured in the presence of the epiphytes. Therefore, we can infer that the presence of *G. usneoides* in nature and the natural release of phenolics by this alga could be affecting the invasive alga *R. okamurae* as high doses of phenolics weaken the photosynthetic system. Thus, we suggest that the management and cultivation of the native alga *G. usneoides* could be an option to reduce the quantity of *R. okamurae* in regions where this invasive alga is dominating and deteriorating the natural ecosystem. At the same time, new experiments evaluating different temperatures, longer exposure times, and additional variables should be conducted to support these findings and obtain a more robust and well-founded understanding of the responses.

## Figures and Tables

**Figure 1 plants-14-02594-f001:**
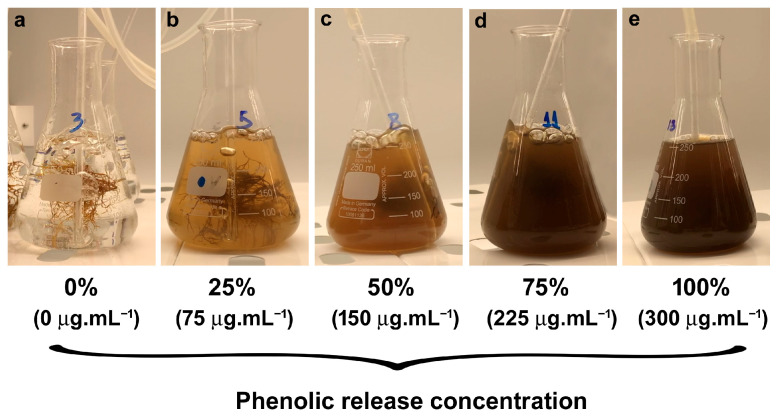
Erlenmeyer flasks contained different concentrations of phenolic compounds in seawater. The 100% concentration (300 µg mL^−1^) was obtained by extracting phenols from *Gongolaria usneoides* thalli using heat, and the other concentrations were prepared by diluting this extract with artificial seawater.

**Figure 2 plants-14-02594-f002:**
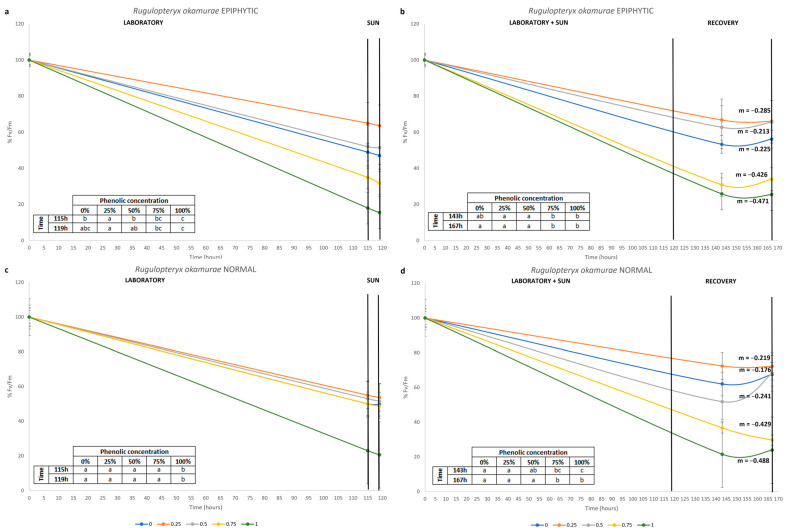
Percentage of *Fv*/*Fm* in epiphytic (**a**) and normal (**c**) *R. okamurae* during 115 h of laboratory and phenolic exposure, 4 h of sun and phenolic exposure, and percentage of *Fv*/*Fm* in epiphytic (**b**) and normal (**d**) *R. okamurae* during 119 h of laboratory + sun and phenolic exposure and 48 h of recovery test, with the slope of the line (m) for each treatment. The tables within the graphs represent the statistical results, where different letters indicate significant differences according to the one-way analysis of variance (phenolic concentration) at each time and Tukey’s test (*p* ≤ 0.05).

**Figure 3 plants-14-02594-f003:**
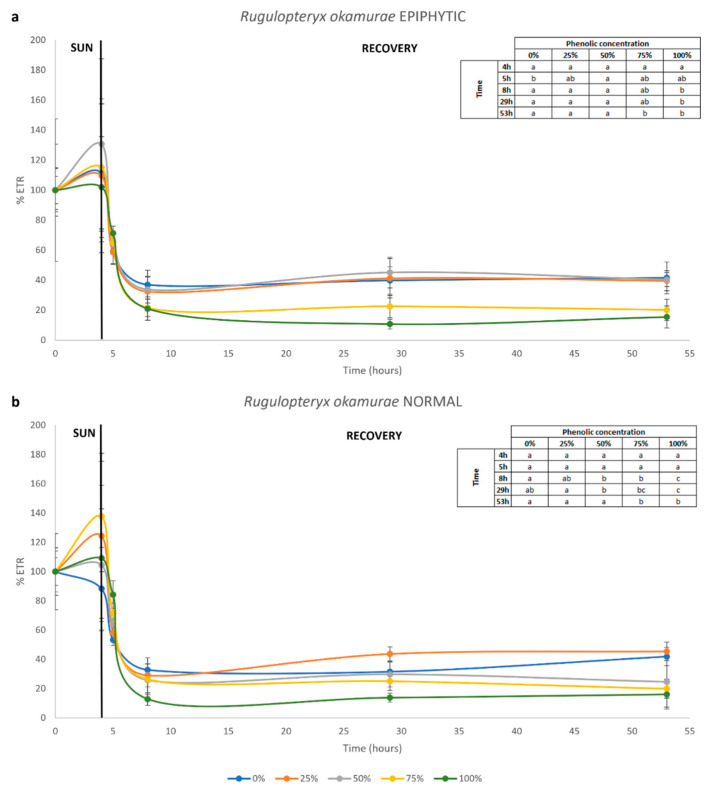
Percentage of ETRs in epiphytic (**a**) and normal (**b**) *R. okamurae* during 4 h of sun and phenolic exposure, and during 48 h of recovery test. The tables within the graphs represent the statistical results; different letters indicate significant differences according to the one-way analysis of variance (phenolic concentration) at each time and Tukey’s test (*p* ≤ 0.05).

**Figure 4 plants-14-02594-f004:**
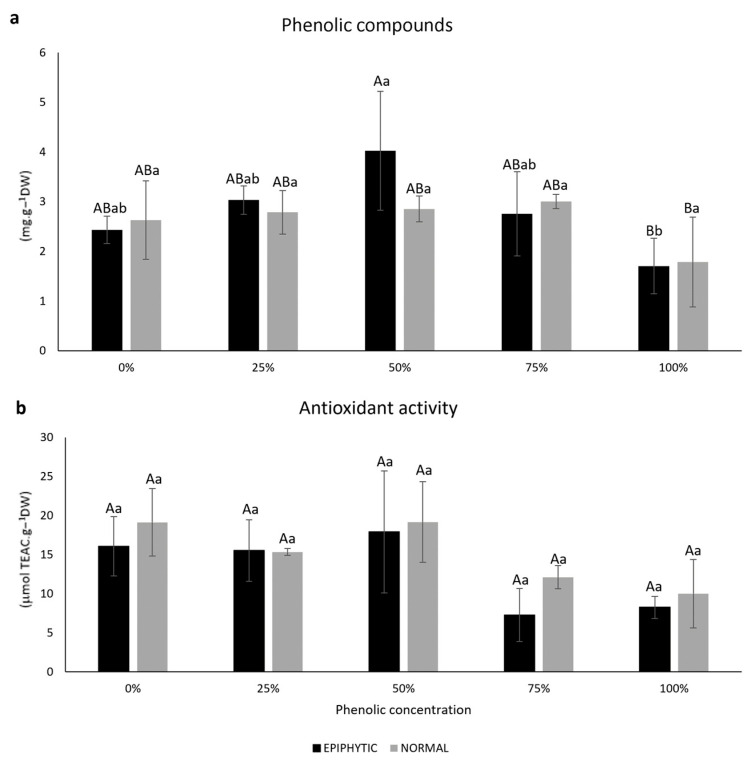
Concentrations of internal phenolics (mg g^−1^ dry weight) (**a**) and concentrations of antioxidant activity (µmol TEAC g^−1^ dry weight) (**b**) in *R. okamurae* ‘epiphytic’ and ‘normal’ at the end of the recovery test (*n* = 3, mean ± SD). Different uppercase letters indicate significant differences according to the two-way analysis of variance (phenolic concentration and *R. okamurae* condition (epiphytic and normal) and Tukey’s test (*p* ≤ 0.05). Different lowercase letters indicate significant differences according to the one-way analysis of variance (concentration of phenolics) and Tukey’s test (*p* ≤ 0.05). No statistically significant difference was observed according to the independent *t*-test analysis (between epiphytic and normal in the same phenolic concentration) (*p* ≤ 0.05).

**Figure 5 plants-14-02594-f005:**
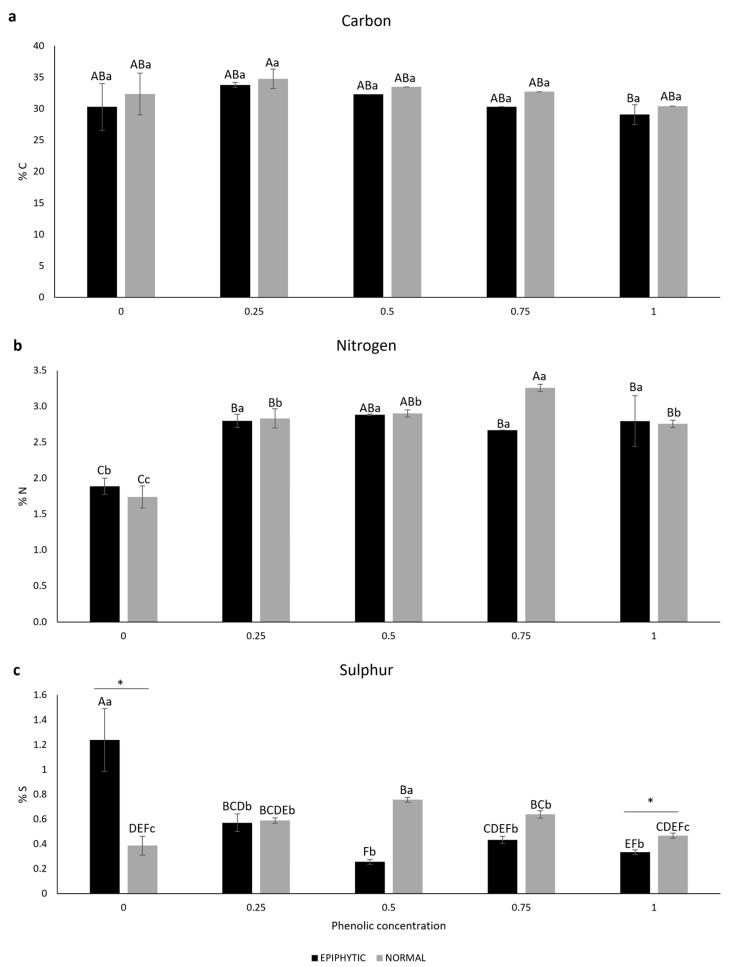
Concentrations of internal carbon (C) (**a**), nitrogen (N) (**b**) and sulfur (S) (**c**) (% dry weight) in *R. okamurae* ‘epiphytic’ and ‘normal’ at the end of the recovery test (*n* = 3, mean ± SD). Different uppercase letters indicate significant differences according to the two-way analysis of variance (phenolic concentration and *R. okamurae* conditions (epiphytic and normal)) and Tukey’s test (*p* ≤ 0.05). Different lowercase letters indicate significant differences according to the one-way analysis of variance (concentration of phenolics) and Tukey’s test (*p* ≤ 0.05). The asterisks indicate statistically significant differences according to the independent *t*-test analysis (between epiphytic and normal in the same phenolic concentration) (*p* ≤ 0.05).

**Figure 6 plants-14-02594-f006:**
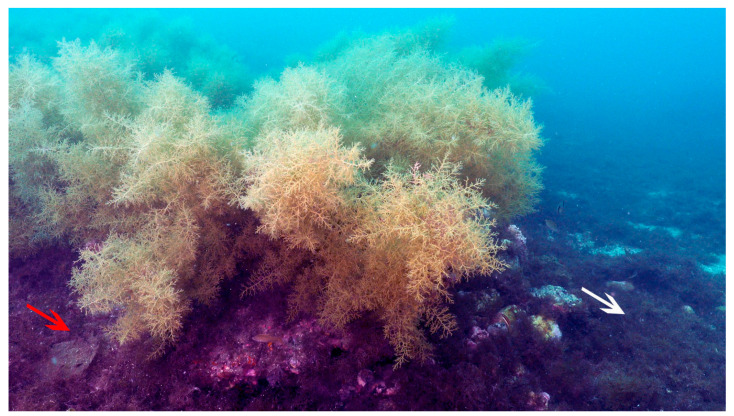
Photo of the collection site. Central presence of *Gongolaria usneoides*, accompanied by *Rugulopterix okamurae* epiphytic (red arrow) and normal (white arrow).

**Figure 7 plants-14-02594-f007:**
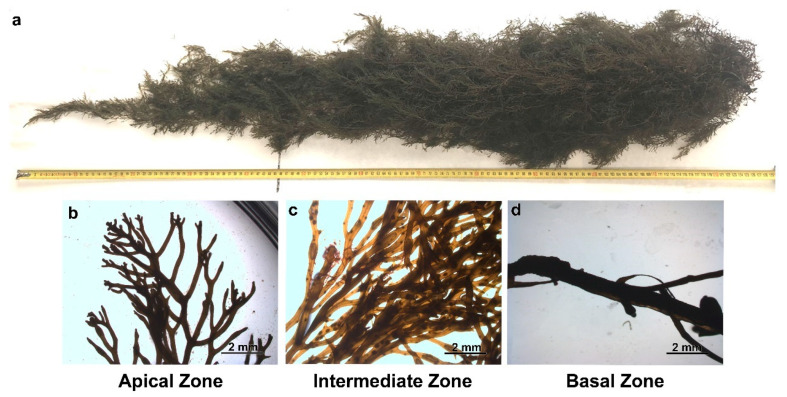
External morphology of *G. usneoides* (**a**). External morphology of each zone of *G. usneoides*: apical (**b**), intermediate (**c**), and basal zone (**d**), observed under a stereoscopic microscope.

**Figure 8 plants-14-02594-f008:**
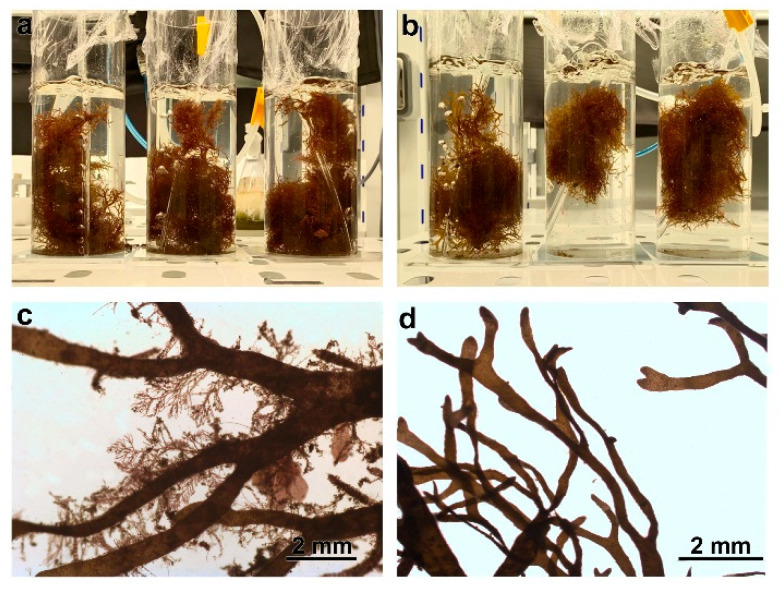
External morphology of epiphytic (**a**) and normal (**b**) *R. okamurae*. External morphology of each *R. okamurae* observed under a stereoscopic microscope: epiphytic (**c**) and normal (**d**).

**Figure 9 plants-14-02594-f009:**
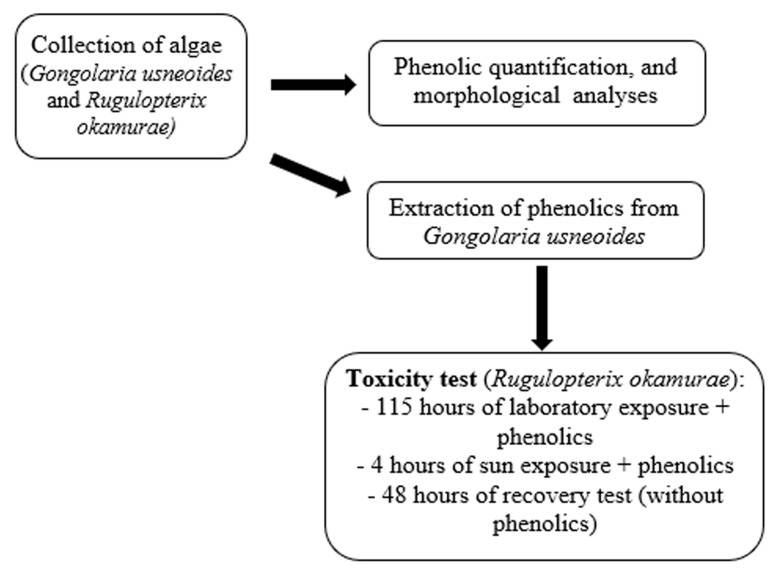
Experimental design with step-by-step from collection to recovery test.

**Table 1 plants-14-02594-t001:** Incident photosynthetically active radiation (PAR) (µmol photons m^−2^ s^−1^) inside the flasks of each phenolic concentration (0, 25, 50, 75 and 100%) in water, ETR in situ and *F_v_*/*F_m_* in epiphytic and normal *R. okamurae* after 4.5 days of laboratory exposure. Different lowercase letters indicate significant differences according to the one-way analysis of variance (concentration of phenolics) and Tukey’s test (*p* ≤ 0.05) for each group (epiphytic and normal) (*p* ≤ 0.05).

	Phenolic Concentration (Water)	Laboratory PAR	In Situ ETR	*Fv*/*Fm*
Epiphytic	0%	190	80.70 ± 6.62 ^a^	0.331 ± 0.03 ^b^
	25%	125	55.08 ± 4.44 ^b^	0.445 ± 0.07 ^a^
	50%	54	23.70 ± 2.96 ^c^	0.347 ± 0.07 ^b^
	75%	45	16.09 ± 4.69 ^c^	0.235 ± 0.04 ^c^
	100%	18	7.19 ± 0.86 ^d^	0.121 ± 0.06 ^d^
Normal	0%	190	78.75 ± 9.41 ^a^	0.367 ± 0.05 ^a^
	25%	125	52.32 ± 6.12 ^b^	0.382 ± 0.05 ^a^
	50%	54	23.87 ± 2.52 ^c^	0.355 ± 0.06 ^a^
	75%	45	14.64 ± 5.43 ^d^	0.337 ± 0.02 ^a^
	100%	18	5.57 ± 2.37 ^e^	0.156 ± 0.02 ^b^

**Table 2 plants-14-02594-t002:** Concentrations of internal carbon to nitrogen ratio (C:N), carbon to sulfur ratio (C:S), and nitrogen to sulfur ratio (N:S) (% dry weight) in *R. okamurae* ‘epiphytic’ and ‘normal’ at the end of the recovery test.

	C:N		C:S		N:S	
**Phenolic** **Concentration**	**Epiphytic**	**Normal**	**Epiphytic**	**Normal**	**Epiphytic**	**Normal**
0%	16.18	18.65	25.36	84.94	1.57	4.64
25%	12.09	12.27	59.76	59.01	4.96	4.81
50%	11.19	11.54	126.10	44.23	11.27	3.82
75%	11.36	10.05	70.03	51.19	6.16	5.10
100%	10.47	11.03	87.37	64.97	8.42	5.89

**Table 3 plants-14-02594-t003:** Concentrations of internal phenolics (mg g^−1^ dry weight) in each zone (apical, intermediate, and basal) of *G. usneoides* and *R. okamurae* ‘epiphytic’ and ‘normal’ from natural (*n* = 3, mean ± SD). Different letters indicate significant differences according to the one-way analysis of variance (zones of *G. usneoides*) and Tukey’s test (*p* ≤ 0.05). Absence of asterisks indicate there is no significant differences according to the independent *t*-test analysis (between *R. okamurae* condition (epiphytic or normal)) (*p* ≤ 0.05).

Species		Internal Phenolic Concentration (mg g^−1^ DW)
*G. usneoides*	Apical	5.98 ± 0.32 ^b^
	Intermediate	6.41 ± 0.98 ^b^
	Basal	14.03 ± 0.90 ^a^
*R. okamurae*	Epiphytic	1.57 ± 0.12
	Normal	1.32 ± 0.11

## Data Availability

All data generated or analyzed during this study are included in this published article.

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
