# Peer review of "Effects of Seawater Polyphenols from Gongolaria usneoides on Photosynthesis and Biochemical Compounds of the Invasive Alien Species Rugulopteryx okamurae (Phaeophyceae, Heterokontophyta)"

_plants, 2025, doi:10.3390/plants14162594_

Round 1

Reviewer 1 Report

Comments and Suggestions for Authors

The present research evaluates the potential of the polyphenols of Gongolaria usneoides to avoid the growth of Rugulopteryx okamurae an alien species.

The research conducted id interesting which could have had interesting consequences for environment.

The introduction is complete with food refeences and states correctly the research objectives

The results are well presented with good figures and tables, and those are well described in the text.

The discussion is quite complete, even if it could be completed with a paragraph indicating the 'real' concentration of polyphenols in see water in the environment close to Gongolaria usneoides, to be able to compare the concentrations studied and the ones that the plant really excrete, because the one which are 'trasnferable' by symbiosis have already be shown.

The material and method section is complete and can be reporduced.

The conclusions reflect the results obtained.

I think the title should be changed as the term "excreted" does not reflect the research conducted as there has an extraction, to evaluate excreted we should analyse the amount of polyphenos excreted my the algae.

Reference 1, is not adapted to the statemetn associated, we need a reference with a larger topic.

The legend of figure 3 should be completed with the condition used to get this colour in the different concentrations of polyphenol

point 2.2.1 the accronyms should be explained when first cited.

Author Response

Referee 1

The present research evaluates the potential of the polyphenols of Gongolaria usneoides to avoid the growth of Rugulopteryx okamurae an alien species. The research conducted id interesting which could have had interesting consequences for environment. The introduction is complete with food references and states correctly the research objectives. The results are well presented with good figures and tables, and those are well described in the text.

The discussion is quite complete, even if it could be completed with a paragraph indicating the 'real' concentration of polyphenols in see water in the environment close to Gongolaria usneoides, to be able to compare the concentrations studied and the ones that the plant really excrete, because the one which are 'transferable' by symbiosis have already be shown.

Response: Due to the complexity of the methodology required to collect water at depth near the algae, we were not able to quantify the concentration of free phenolic compounds in deep water. However, under controlled laboratory conditions (temperature and PAR), we achieved the natural release of phenols approximately 150 µg mL⁻¹ after 2 days, and the natural release in natural tide pools reached approximately 18-50 µg mL⁻¹ of phenolics after 2 h of sun exposure. We have added this information to the text.

The material and method section is complete and can be reproduced.

The conclusions reflect the results obtained.

I think the title should be changed as the term "excreted" does not reflect the research conducted as there has an extraction, to evaluate excreted we should analyse the amount of polyphenols excreted by the algae.

Response: We agree and have changed the title.

Reference 1, is not adapted to the statement associated, we need a reference with a larger topic.

Response: We agree and have changed the reference.

The legend of figure 3 should be completed with the condition used to get this colour in the different concentrations of polyphenol.

Response: We agree and have changed the caption.

Point 2.2.1 the acronyms should be explained when first cited.

Response: We agree and have explained the acronyms.

Reviewer 2 Report

Comments and Suggestions for Authors

This manuscript aims to analyze the impact of phenolic compounds released by Gongolaria usneoides, a native macroalga, on the photosynthetic and biochemical behavior of Rugulopteryx okamurae.  The study addresses a relevant topic within the context of biological invasions, particularly in European coastal ecosystems. Overall, the manuscript is well written, with no major issues, but some sections need revising for clarity.

The Abstract lacks a concise description of the experiment. It should include a brief description of the phenolic concentrations, time points and measured traits.

The Introduction is brief and and would benefit from incorporating the ecological importance of phenolic compounds and their potential applications in managing invasive macroalga. Additionally, the literature review should be updated to include more recent studies. For example, references 28 and 29 (line 61) relate to studies from 1964 and 1978 respectively. The inclusion of reference [39] at the end of the section should be reconsidered.

In the Results section, Figures 4 and Figure 5 must be reformatted for better clarity. The phenolic content was measured at the beginning and after 115 hours of laboratory exposure to phenolic compounds. Then, after four hours of exposure to sunlight and phenolic compounds, the content was measured again. After this period, the recovery phase began, with determinations taken at 1, 3, 24 and 48 hours after the start of the phase. This should be clearly represented in both figures. In addition, connecting lines between 0 and 115 hours seem inappropriate since they denote continuous change, which cannot be inferred from the results. Consider plotting these time points distinctly. Figures 4b and 4d should be revised to accommodate the information expressed in lines 416–417. Additionally, the inclusion of the slope (m) should be explained in the 'Materials and Methods' section. The lowercase letters in Figure 4 are difficult to visualize and should be enlarged.

The Discussion section should also be revised. Some statements are not directly supported by the data. Some statements are not directly supported by the data. For example, in lines 296–298 it is concluded that the large production of phenolic compounds by G. usneoides could be viewed as a defence and protection mechanism, especially under stress. However, no specific stressor was studied, and the incorporated references are about a different species. The same occurred in lines 361–363, where some antioxidant enzymes were mentioned but not determined in the present study. Therefore, the discussion should focus on the results presented.

The Material and Methods section should be more detailed. The criteria for selecting the phenolic concentrations should be clearly explained (lines 400), given that the natural phenolic concentration ranges from 18 to 58 µg/mol (line 117). Explain why 115 hours was chosen for exposure and 48 hours for recovery. The methodology for calculating the slope of the lines used in the figures should be explained.

The Conclusion section includes several general statements that are not supported by the data. For example, it states that benzoic acid is the most predominant phenolic compound (line 500) which gives this species higher antioxidant activity than R. okamurae. However, this was not directly determined in this study. Therefore, these general assumptions should be interpreted with caution. The conclusion should focus on the practical application of these findings, e.g. how to manipulate the phenolic content of G. usneoids, and analyse the ecological effects of a high phenolic content in G. usneoids itself and in other algae species. It should also suggest further studies under different environmental conditions (e.g. different temperatures and longer exposure times) to validate these findings.

Other comments

Line 111: The sentence “Absence of asterisks indicate there is no significant differences according to the independent t-test analysis (between R. okamurae condition (epiphytic or normal) (p ≤ 0.05).”should be revised for clarity.

Line 149: Add a note explaining the lowercase letters in the table.

Line 152: The lowercase letters are missing in a).

Line 248: Clarify the meaning of “*” in c).

Line 316: Confirm if the exposure time is “2 hours” or “4 hours”.

Line 346-354: Revise for consistency in the presentation of R. okamurae.

Line 387 and 445: Correct the temperature units.

Line 423: Convert “4.5 days” to hours for consistency with the other time points.

Line 489-490: Revise “lower” in the sentence. I don't think that is correct.

Author Response

Referee 2 

This manuscript aims to analyze the impact of phenolic compounds released by Gongolaria usneoides, a native macroalga, on the photosynthetic and biochemical behavior of Rugulopteryx okamurae.  The study addresses a relevant topic within the context of biological invasions, particularly in European coastal ecosystems. Overall, the manuscript is well written, with no major issues, but some sections need revising for clarity.

The Abstract lacks a concise description of the experiment. It should include a brief description of the phenolic concentrations, time points and measured traits.

Response: We agree and have added some information in the abstract.

The Introduction is brief and would benefit from incorporating the ecological importance of phenolic compounds and their potential applications in managing invasive macroalga. Additionally, the literature review should be updated to include more recent studies. For example, references 28 and 29 (line 61) relate to studies from 1964 and 1978 respectively. The inclusion of reference [39] at the end of the section should be reconsidered.

Response: We agree and have added new information to the introduction. While we have kept the original references as they provide a foundational basis, we have also included new and updated references.

In the Results section, Figures 4 and Figure 5 must be reformatted for better clarity. The phenolic content was measured at the beginning and after 115 hours of laboratory exposure to phenolic compounds. Then, after four hours of exposure to sunlight and phenolic compounds, the content was measured again. After this period, the recovery phase began, with determinations taken at 1, 3, 24 and 48 hours after the start of the phase. This should be clearly represented in both figures. In addition, connecting lines between 0 and 115 hours seem inappropriate since they denote continuous change, which cannot be inferred from the results. Consider plotting these time points distinctly. Figures 4b and 4d should be revised to accommodate the information expressed in lines 416–417. Additionally, the inclusion of the slope (m) should be explained in the 'Materials and Methods' section. The lowercase letters in Figure 4 are difficult to visualize and should be enlarged.

Response: We agree that the font size in Figure 4 was too small, and we have enlarged it for better readability. We have also added a detailed explanation of the slope (m) in the Materials and Methods section, as suggested. Regarding the graphical format, we appreciate your comment; however, we believe that the current figure design is the most effective way to represent the temporal dynamics of the experiment. Representing the data as histograms or in table would make it more difficult to visualize the progression and comparison of phenolic content over time. Our goal was to highlight changes between experimental phases, and we feel this format better supports that objective.

The Discussion section should also be revised. Some statements are not directly supported by the data. Some statements are not directly supported by the data. For example, in lines 296–298 it is concluded that the large production of phenolic compounds by G. usneoides could be viewed as a defence and protection mechanism, especially under stress. However, no specific stressor was studied, and the incorporated references are about a different species. The same occurred in lines 361–363, where some antioxidant enzymes were mentioned but not determined in the present study. Therefore, the discussion should focus on the results presented.

Response: We agree and have changed these parts in the discussion section.

The Material and Methods section should be more detailed. The criteria for selecting the phenolic concentrations should be clearly explained (lines 400), given that the natural phenolic concentration ranges from 18 to 58 µg/mol (line 117). Explain why 115 hours was chosen for exposure and 48 hours for recovery. The methodology for calculating the slope of the lines used in the figures should be explained.

Response: We agree and following the suggestions of the other reviewers, we have added this information about the concentration in the results section 2.1. The explanation abou the time was added in material and methods, section 4.4. And the explanation about the slope was added in material and methods, section 4.5.

The Conclusion section includes several general statements that are not supported by the data. For example, it states that benzoic acid is the most predominant phenolic compound (line 500) which gives this species higher antioxidant activity than R. okamurae. However, this was not directly determined in this study. Therefore, these general assumptions should be interpreted with caution. The conclusion should focus on the practical application of these findings, e.g. how to manipulate the phenolic content of G. usneoids, and analyse the ecological effects of a high phenolic content in G. usneoids itself and in other algae species. It should also suggest further studies under different environmental conditions (e.g. different temperatures and longer exposure times) to validate these findings.

Response: We agree and as we do not have the characterization of phenolics, we eliminate the benzoic acid from conclusion, even knowing by literature that this is the most abundant phenolic in Gongolaria. We agree and we have added the suggestions for new experiments.

Other comments

Line 111: The sentence “Absence of asterisks indicate there is no significant differences according to the independent t-test analysis (between R. okamurae condition (epiphytic or normal) (p ≤ 0.05).”should be revised for clarity.

Response: We have checked, and the information is correct.

Line 149: Add a note explaining the lowercase letters in the table.

Response: We agree and have added this information.

Line 152: The lowercase letters are missing in a).

Response: We agree and have corrected the figure.

Line 248: Clarify the meaning of “*” in c).

Response: We agree and have clarified in the text.

Line 316: Confirm if the exposure time is “2 hours” or “4 hours”.

Response: We have checked and the correct is 2 hours.

Line 346-354: Revise for consistency in the presentation of R. okamurae.

Response: We agree and have changed throughout the text.

Line 387 and 445: Correct the temperature units.

Response: We agree and have changed in the text.

Line 423: Convert “4.5 days” to hours for consistency with the other time points.

Response: We agree and have changed the value in the figure.

Line 489-490: Revise “lower” in the sentence. I don't think that is correct.

Response: We agree and have changed the sentence.

Reviewer 3 Report

Comments and Suggestions for Authors

The manuscript entitled: ‘Effects of Excreted Polyphenols from Gongolaria usneoides on Photosynthesis and Biochemical Compounds of the Invasive Alien Species Rugulopteryx okamurae (Phaeophyceae, Heterokontophyta)’ is a comprehensive research on allelopathy. In my opinion, such a detailed research will certainly be of great interest to researchers around the world as the study addresses the highly important topic of the  Allelopathic interaction between native and invasive brown alga. The manuscript is suitable for publication in the journal after minor revision.

My comments:

  1. Keywords: add “Photosynthesis”

  1. Ressults 1. Characterization of Fresh Biological Material. This section is more suitable to be placed in “Materials and Methods”.

  1. Are the tested concentrations of the total phenols (up to 300 µ g mL-1 )ecologically realistic? How do these concentrations compare with levels that might occur naturally in marine under conditions where Gongolaria usneoides and Rugulopteryx okamurae grow?

  1. Which phenolicin the total phenols played the main toxicity to the invasive brown alga? So more data such as LC-MS is needed.

  1. The quality of figure9 should be improved.

Author Response

 Referee 3

  1. Keywords: add “Photosynthesis”

Response: We agree and added the work as keyword.

  1. Ressults 1. Characterization of Fresh Biological Material. This section is more suitable to be placed in “Materials and Methods”.

Response: We agree and changed this part of the text to material and methods section.

  1. Are the tested concentrations of the total phenols (up to 300 µg mL-1) ecologically realistic? How do these concentrations compare with levels that might occur naturally in marine under conditions where Gongolaria usneoidesand Rugulopteryx okamurae grow?

Response: Under controlled laboratory conditions (temperature and PAR), we achieved approximately 150 µg mL⁻¹ release of phenols. However, since the objective of this study was to test the toxicity of phenols from Gongolaria to R. okamurae, and the laboratory conditions were limited in promoting phenol release, we decided to extract these compounds in order to work with a broader range of concentrations than those naturally released. And, due to the complexity of the methodology required to collect water at depth near the algae, we were not able to quantify the concentration of free phenolic compounds in deep water. We have added this information to the text.

  1. Which phenolic in the total phenols played the main toxicity to the invasive brown alga? So more data such as LC-MS is needed.

Response: In the present study, we did not perform LC-MS analysis of phenolic compounds. Instead, we based our discussion on previously published references that report the phenolic composition of each alga. As mentioned in the discussion section, the major phenolic compounds differ between species, for example, in the case of Gongolaria, the most abundant phenolic compound is benzoic acid [46], whereas in R. okamurae, benzoic acid is not present in its composition, and the predominant phenolic compound is gallic acid.

  1. The quality of figure 9 should be improved.

Response: We agree and improved the quality of figure 9.